# A Macro-meso damage coupling rock mass damage model based on improved internal crack analysis

Haian Liang[1], Miao He [1]*, Hongliang Zhao[1], Yiru Zhang[1], Qingrui Lu[1], Shuren Hao[1], Zhiying Gong[1], Hao Guan[2]

**1** School of Civil and Architectural Engineering, East China University of Technology, Nanchang, Jiangxi, China, **2** Jiangxi Provincial Architectural Design and Research Institute Group Co. Ltd. Nanchang, Jiangxi, China

* 1109423900@qq.com

## Abstract

The complex geological genesis and different geological environments of rocks can lead to defects such as micropores, microcracks, and joints, resulting in the degradation of their engineering mechanical properties. In order to describe the deformation and failure mechanisms of jointed rock, a rock damage constitutive model that accounts for the geometric parameters and mechanical properties of internal joints within the rock is proposed. By leveraging the principles of damage mechanics and the Lemaitre strain equivalence hypothesis, macroscopic and microscopic damages in rock were combined, ultimately resulting in the development of a damage constitutive model that encompassed both scales of damage. The damage constitutive model was verified by uniaxial compression tests of clay-like rock under different joint dip angles and joint area combinations. The results indicated that the proposed damage constitutive model had clear physical significance and could fully reflect the process of rock failure, which was highly consistent with experimental results. The model parameter $m$ reflected the brittleness of jointed rocks, while $F_0$ reflected the average strength of jointed rocks. The combination of macroscopic and microscopic analysis methods used in this study is reasonable, and the established damage constitutive model reflects the mechanical behavior of rocks and fits the experimental results well.

## 1. Introduction

The formation process of rocks results in micro-cracks, joints, and other macro- and micro-scale defects [1–4]. These defects at different scales influence the mechanical properties of engineering rock masses in various ways.

The constitutive relationship of such defective rock masses has persistently been a key area of interest in geotechnical engineering [5–10]. Krajcinovic D et al [11]. were the first to introduce the Weibull function to describe the random distribution characteristics

**Data availability statement:** All relevant data are within the manuscript.

**Funding:** the National Natural Science Foundation of China (Grant No. 42077255); the National Natural Science Foundation of China (Grant No. 42002258).

**Competing interests:** The authors declare no competing interests.

of rock's micro-element strength, opening up a constitutive model that can better reflect the rock's stress-strain curve and providing new ideas for subsequent research. Subsequently, numerous scholars further studied the damage constitutive model from different perspectives based on this foundation [12–16]. Chen et al [17]. Based on statistical damage theory and Mohr Coulomb strength criterion, a logistic model is used to derive the damage evolution equation, and a statistical damage constitutive model is established to describe the entire process of rock failure. Ren et al. [18]. Considering the relationship between damage evolution and plastic flow, a rock damage model considering plastic strain is proposed by introducing two internal variables: damage and plasticity. Guo et al [19]. By utilizing the Usher function and considering the nonlinear deformation characteristics of rocks during the initial compaction stage, a new and more applicable damage constitutive model was established by combining damage mechanics theory with effective medium theory. These studies have greatly promoted the development of statistical damage theory, but these studies only considered the microscopic damage in rocks. Some scholars have studied the damage caused by joints in rocks and defined damage tensors to represent the geometric shape of joints [20,21], However, due to the inherent mechanical properties such as internal friction angle of joints, Chen et al [22]. established a rock mass constitutive model considering crack closure and surface friction effects in jointed rock masses based on the energy equivalence principle. The above study only considered the influence of macroscopic joints in rocks and ignored microscopic damage. In fact, considering only macroscopic or microscopic damage is not appropriate, and both need to be considered together [23–25], Liu et al [26] proposed a damage constitutive model that describes the deformation and strength characteristics of discontinuous jointed rocks under uniaxial cyclic compression, taking into account the geometric parameters and mechanical properties of discontinuous joints. Zhang et al [27]. proposed a rock damage constitutive model that considers the geometric parameters and mechanical properties of joints to describe the deformation and failure mechanism of jointed rock masses. Liu et al [28,29]. established a damage constitutive model for non-persistent joints under dynamic loading by taking into account macroscopic defects such as joints and microscopic damage in rocks. Zhao et al [30]. calculated the damage tensor considering the anisotropy of joints, thereby establishing a jointed rock mass damage constitutive model based on the coupling of macro- and micro-defects, which is subsequently verified through experiments. Yuan et al [31]. established a macro-meso damage constitutive model for non-intersecting fractures based on the axial strain strength theory. From the current research status, it can be seen that although extensive research has been conducted on the constitutive behavior of rock damage, and progress has been made in establishing statistical damage constitutive models and damage constitutive models under the coupling of macroscopic and microscopic defects. However, the existing damage constitutive models partially consider micro damage and macro damage separately, without taking into account the coupling of the two. The damage constitutive equation established by coupling two types of damage mostly studies discontinuous central cracks. Considering the multi-scale damage and different spatial distributions of joints in rocks, there is still room for further improvement based on existing research.

In view of the shortcomings of the above research, this study established a damage constitutive model based on fracture mechanics that includes macroscopic microscopic coupling effects of internal fractured rocks. This model considers that the strength of rock micro units follows a statistical distribution and includes the concept of the generation and evolution of rock damage under macroscopic microscopic coupling effects. In addition, it is based on the Drucker Prager (D-P) strength criterion. This study is a further supplement to the macroscopic and microscopic defect constitutive model system.

## 2. Establishment of a damage constitutive model

The derivation of macro-micro coupling damage variables is carried out from two aspects: macro damage variables and micro damage variables, as shown in Fig 1.

The damage in rocks containing joint fissures is categorized by Yuan et al [31]. into two types: mesoscopic damage, which is generated during the loading process and represented by $D_I$, and macroscopic damage, which is caused by preformed joints and denoted as $D_{II}$. The coupling of these two types of damage is termed the macro-mesoscopic coupled damage, represented by $D_{III}$. The expression for $D_{III}$ is formulated as follows [32]:

$$D_{III} = 1 - \frac{(1 - D_I)(1 - D_{II})}{1 - D_I D_{II}}$$

(1)

where: $D_I$ is the meso damage of intact rock caused by loading, $D_{II}$ is the damage caused by prefabricated joints, $D_{III}$ is the macro-meso coupling damage variable.

Under the action of axial compressive load $\sigma_1$, the damage strain energy release rate $U$ of a rock containing cracks is expressed as [20]:

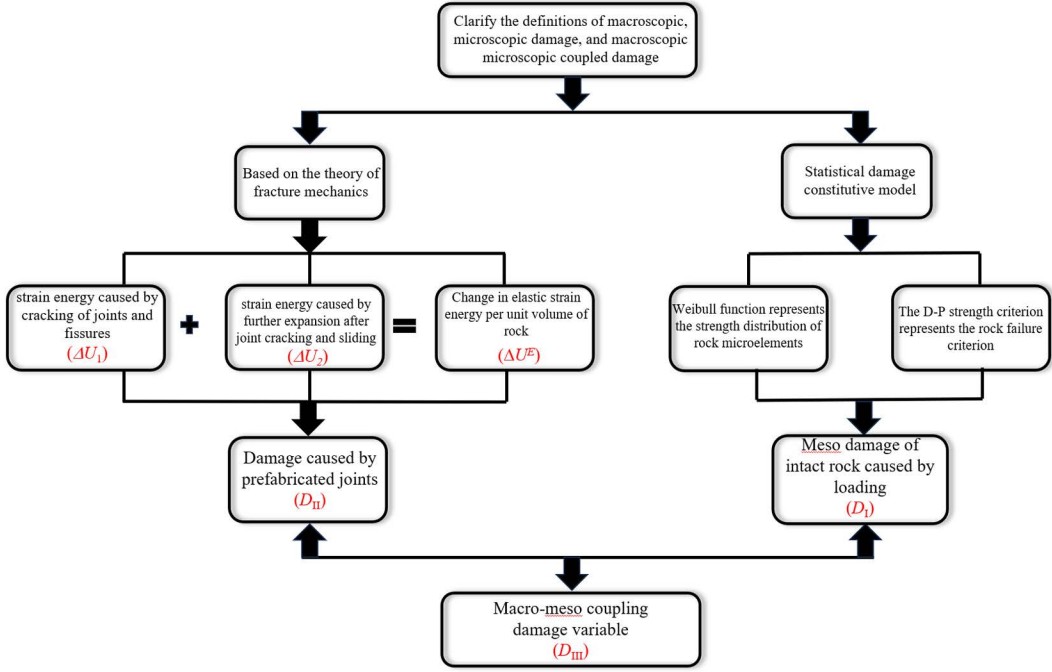

**Fig 1. Deduction process of damage variables.**

$$U = -\sigma_1 \Big/ 2E(1-D_{III})^2 \qquad (2)$$

where: $U$ is the release rate of damage strain energy, $E$ is the modulus of elasticity.

Under the action of axial compressive load $\sigma_1$, the elastic strain energy per unit volume is represented by $U^E$. The elastic strain energy in this state can be expressed as [20]:

$$U^E = -(1-D_{III})U \qquad (3)$$

By combining Equation 2 and Equation 3, we obtain:

$$U^E = \sigma_1^2 \Big/ 2E(1-D_{III}) \qquad (4)$$

When there are no macroscopic damages in the rock, $D_{II}=0$, and the microscopic damages are equal to the macro-meso coupled damages, denoted as $D_I = D_{III}$. At this point, the strain energy per unit volume is:

$$U_1^E = \sigma_1^2 \Big/ 2E(1-D_I) = \sigma_1^2 \Big/ 2E(1-D_{III}) \qquad (5)$$

By combining Equations 4 and 5, we can obtain the change in elastic strain energy per unit volume of rock caused by macroscopic damage as:

$$\Delta U^E = U^E - U_1^E = \sigma_1^2 \Big/ 2E(1-D_{III}) - \sigma_1^2 \Big/ 2E(1-D_I) \qquad (6)$$

Assuming the volume of the loading test is $V$, and considering the presence of joints in the elastic body, the change in elastic strain energy due to these joints is given by:

$$\Delta U^E = V\left[\frac{\sigma_1^2}{2E(1-D_{III})} - \frac{\sigma_1^2}{2E(1-D_I)}\right] \qquad (7)$$

According to fracture mechanics theory, the strain energy caused by cracking of joints and fissures in rock mass $\Delta U_1$ is [20]:

$$\Delta U_1 = \int_0^A GdA = \left[(1-\nu^2)/E\right]\int_0^A \left(K_I^2 + K_{II}^2\right)dA \qquad (8)$$

where: $G$ is the crack surface release rate, $\nu$ is the Poisson's ratio, $A$ is the surface area of the crack, $K_I$ is the mode I intensity factor of wing crack at the crack tip, $K_{II}$ is the mode II intensity factor of wing crack at the crack tip.

Under the action of load, the joint will further expand due to shear slip, and the strain energy caused by crack is $\Delta U_2$ [33]:

$$\Delta U_2 = 2\int_0^l Gdl = \left([1+k_0](1+\nu)/2E\right)\int_0^l \left(K_I^2 + K_{II}^2\right)dl \qquad (9)$$

where: $l = \left(2\sqrt{2}a - 2a\sin\alpha\right)\big/\sin(\theta+\alpha)$, $k_0 = (3-\nu)/(1+\nu)$, Since the propagation direction of the wing-shaped crack is parallel to the direction of the applied stress when it ultimately fails, then $\theta + \alpha = \pi/2$ [33,34].

Given that $\Delta U_1$, $\Delta U_2$, and $\Delta U^E$ are all changes in elastic strain energy due to macroscopic damages within the rock, where $\Delta U_1$ plus $\Delta U_2$ equals $\Delta U^E$. Therefore, by combining equations 7, 8, and 9, the damage variable caused by macroscopic damage can be determined:

$$D'_{II} = 1 - \frac{1}{1 + \frac{2-2\nu^2}{V\sigma^2} \int_0^A (K_I^2 + K_{II}^2)dA + \frac{(1+k_0)(1+\nu)}{V\sigma^2} \int_0^l (K_I^2 + K_{II}^2)dl} \tag{10}$$

As can be analyzed from Equation 10, the expression of macroscopic damage variables obtained is related to the strength factor at the joint tip and the physical and mechanical parameters of the rock itself. And the accuracy in calculating the macroscopic damage variable depends critically on the selection of the stress intensity factor. Cracks can be classified into through-thickness cracks, surface cracks, and internal cracks based on their location. Depending on the crack location, the calculation methods and manifestations of the crack intensity factor vary accordingly.

This study focuses on the inclined internal joints in rocks as the research object, for the analysis of internal cracks, it is commonly done to place the crack within a three-dimensional infinite body and simplify the crack surface into an elliptical or circular shape. Under the action of axial compressive load $\sigma_1$, the crack propagation at the tip of the joint wing is shown in Fig 2.

The normal stress and shear stress on the joint surface are as follows:

$$\begin{cases} \sigma_n = \sigma_1 \cos^2 \alpha \\ \tau_n = \sigma_1 \sin 2\alpha / 2 \end{cases} \tag{11}$$

where: $\sigma_n$ is the normal stress, $\tau_n$ is the shear stress, $\alpha$ is the dip angle of fracture, $\sigma_1$ is the Axial pressure, $a$ is the half length of the crack, $l$ is the extension length of wing crack, $\theta$ is the starting angle of wing crack.

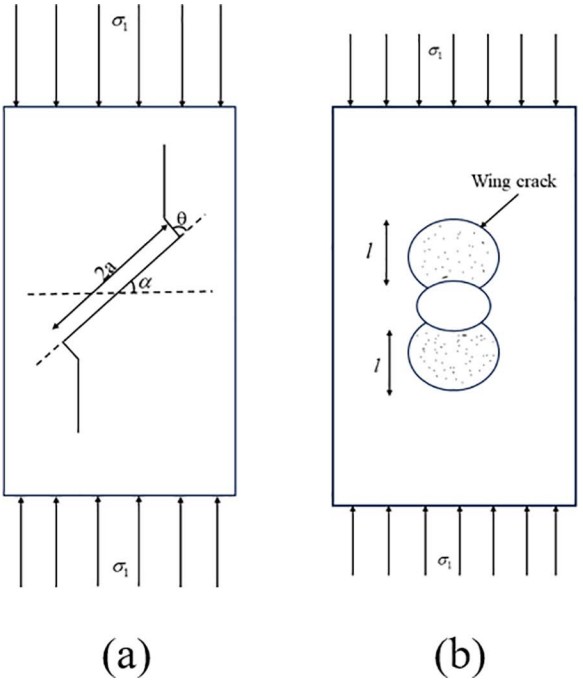

(a)　　　　　　(b)

**Fig 2. Schematic diagram of wing crack propagation at crack tip(a) Schematic diagram of wing crack growth model  (b) Schematic diagram of wing crack propagation.**

The friction angle of the rock joint surface is assumed to be $\phi$, resulting in the coefficient of friction for the joint surface being defined as $\mu = \tan(\phi)$. When external loads are applied to the rock, if the shear force exceeds the friction provided by the joint surface, failure along the joint surface is undergone by the rock; otherwise, it remains intact. Thus, this can be defined as the force causing structural plane slipping.

$$\tau_f = \begin{cases} 0 - - - - - -\tan\phi > \tan\alpha \\ \sigma_1\sin 2\alpha/2 - \mu\sigma_1\cos^2\alpha - - - - - -\tan\phi \leq \tan\alpha \end{cases} \tag{12}$$

where: $\phi$ is the friction angle of the structural plane, $\mu$ is the coefficient of friction

According to the research findings of ASHBY et al [35], the intensity factors $K_I$ and $K_{II}$ corresponding to the crack tip of an internal crack are expressed as follows:

$$K_I = \frac{\pi a^2 \tau_f \sin\theta}{\sqrt{(\pi(l + \lambda a))^3}} + \frac{\pi}{2}\left(\sigma_3 - \sigma_3^i\right)\sqrt{\pi l} \tag{13}$$

$$K_{II} = -\frac{\pi a^2 \tau_f \cos\theta}{\sqrt{\pi(l + \lambda a)^3}} - \frac{\pi}{2}\left(\sigma_3 - \sigma_3^i\right)\sqrt{\pi l} \tag{14}$$

where: $\lambda$ is the correction factor, $\sigma_3$ is the confining pressure, $\sigma_3^i$ is the internal stress.

It can be observed that when the parameter $l$ is greater than 0, the stress intensity factor for the wing crack can be adequately expressed. However, when $l=0$, the stress intensity factor tends to infinity, which is inconsistent with actual conditions. Therefore, the term $\lambda a$ is introduced to resolve this issue.

When the crack is at its cracking limit, where $l=0$, the aforementioned formula can be rewritten as follows:

$$K_I = \frac{\pi a^2 \tau_f \sin\theta}{\sqrt{(\pi\lambda a)^3}} \tag{15}$$

$$K_{II} = \frac{\pi a^2 \tau_f \cos\theta}{\sqrt{(\pi\lambda a)^3}} \tag{16}$$

The stress intensity factor previously discussed is derived under the assumption that the crack is embedded in an infinitely large matrix. However, when the sample size is relatively small, the dimensions of the sample need to be taken into account. Therefore, based on the research findings of Isida [36], the stress intensity factor can be modified as follows:

$$K_I = \frac{\pi a^2 \tau_f \sin\theta}{\sqrt{(\pi\lambda a)^3}}\sqrt{\sec\left(\frac{\pi a}{w}\right)} \tag{17}$$

$$K_{II} = \frac{\pi a^2 \tau_f \cos\theta}{\sqrt{(\pi\lambda a)^3}}\sqrt{\sec\left(\frac{\pi a}{w}\right)} \tag{18}$$

where: $W$ is the Flat width.

By substituting Equations [12], [17] and [18] into Equation [10], we can obtain the expression for the macroscopic damage variable that characterizes the internal cracks:

$$D'_{II} = 1 - \frac{1}{1 + \frac{2\pi^2(1-\nu^2)a^4\cos^2\alpha(\sin\alpha-\tan\phi\cos\alpha)^2}{V(\pi\lambda a)^3}\int_0^A \sec(\frac{\pi a}{w})dA + \frac{(1+k_0)(1+\nu)\pi^2 a^4\cos^2\alpha(\sin\alpha-\tan\phi\cos\alpha)^2}{V(\pi\lambda a)^3}\int_0^l \sec(\frac{\pi a}{w})dl} \tag{19}$$

Yuan et al [31]. also presented a macroscopic damage expression for non-penetrating cracks and provided a detailed derivation of the calculation problem for the coupling of two types of damage as well as the macroscopic and mesoscopic damage constitutive expressions. The specific formulas are as follows:

Macroscopic Damage Expression:

$$D''_{II} = 1 - \frac{1}{1 + \frac{8(1-\nu^2)a^2\sigma^2\cos^2\alpha(\sin\alpha-\tan\phi\cos\alpha)^2}{V\pi l^*}\int_0^A \sec\left(\frac{\pi a}{w}\right)dA} \tag{20}$$

where: $l^* = 0.27a$

The constitutive model expression incorporating both macroscopic and mesoscopic damages:

$$\sigma = E_0\varepsilon\left(1 - D_{III}\right) = E_0\varepsilon\frac{\left(1 - D_I\right)\left(1 - D_{II}\right)}{1 - D_I D_{II}} \tag{21}$$

where: $E_0$ is the elastic modulus of intact rock.

## 3. Verification and parameter analysis of a uniaxial compression constitutive model considering both macroscopic and mesoscopic damages

### 3.1 Uniaxial compression test

Cement, water, standard sand and barite powder are used to mix and pour the complete clay rock at the mass ratio of 1:0.5:0.5:1.17. The specific pouring process is shown in Fig 3. The clay like rocks with different joint dip angles are prepared by using the independently designed mold. The prefabricated joints are made of 0.18 mm thick polypropylene film, which is primarily composed of polyethylene resin and natural stone powder. The size of the joints is achieved by changing the area of polyethylene film, which is called the proportion of joint area in this study. Finally, the complete clay rock and the jointed clay rock with 15°and 35°dip angles and 50% and 75% of the joint area are prepared. The pouring size of the sample is shown in Fig 4.

The uniaxial loading method is adopted. Firstly, 1kN force is applied for preloading. The purpose is to make the sample fully contact with the instrument, and then the test is carried out at the axial loading rate of 0.5 kN·s⁻¹ until the test is completely destroyed. The test results are shown in Fig 5.

As shown in Fig 5, the mechanical properties of the specimens are observed to vary with the joint dip angle and joint density, accompanied by distinct failure patterns under different conditions. Intact specimens exhibit tensile failure, which represents a typical uniaxial compressive failure mode. In contrast, specimens with 15°-50% and 35°-50% joint configurations demonstrate mixed tensile and shear cracks on their surfaces, with both crack types occurring in comparable quantities. Specimens with 15°—75% and 35°—75% joint configurations, however, display predominantly tensile cracks accompanied by a limited number of shear cracks. This contrast highlights the transition from combined tensile-shear failure mechanisms to tensile-dominated failure modes as joint density increases under fixed dip angles.

### 3.2 Determination of the Correction Coefficient λ

Scholars such as Yuan directly introduce an effective length $l^*$=0.27 a to address the singularity of the stress intensity factor at the tip [31,37], which corresponds to $\lambda$=0.27 in this paper. However, current research indicates that the effective length is not

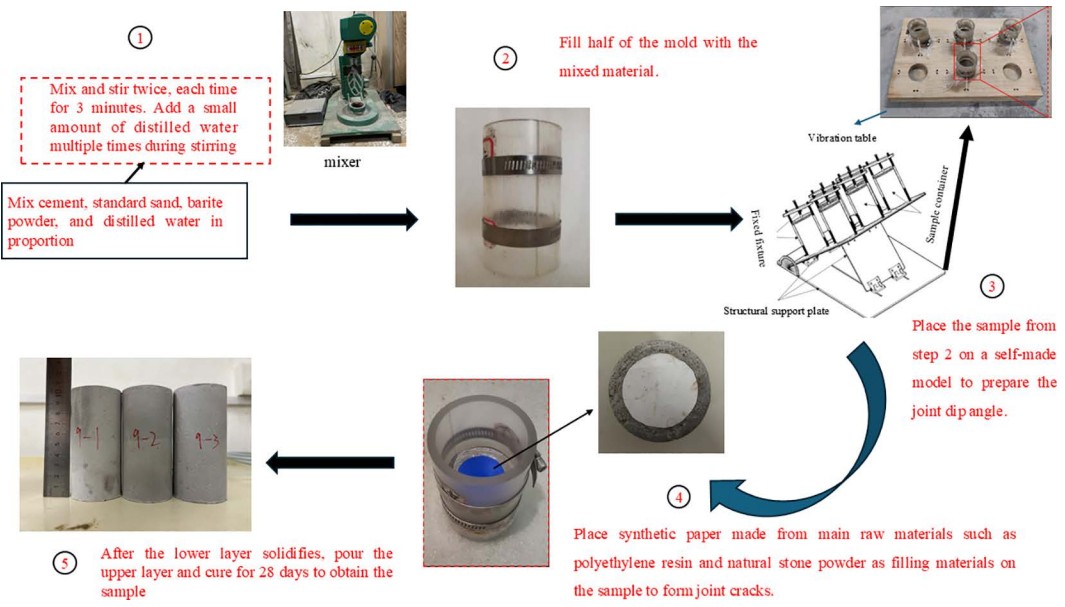

**Fig 3. Pouring process.**

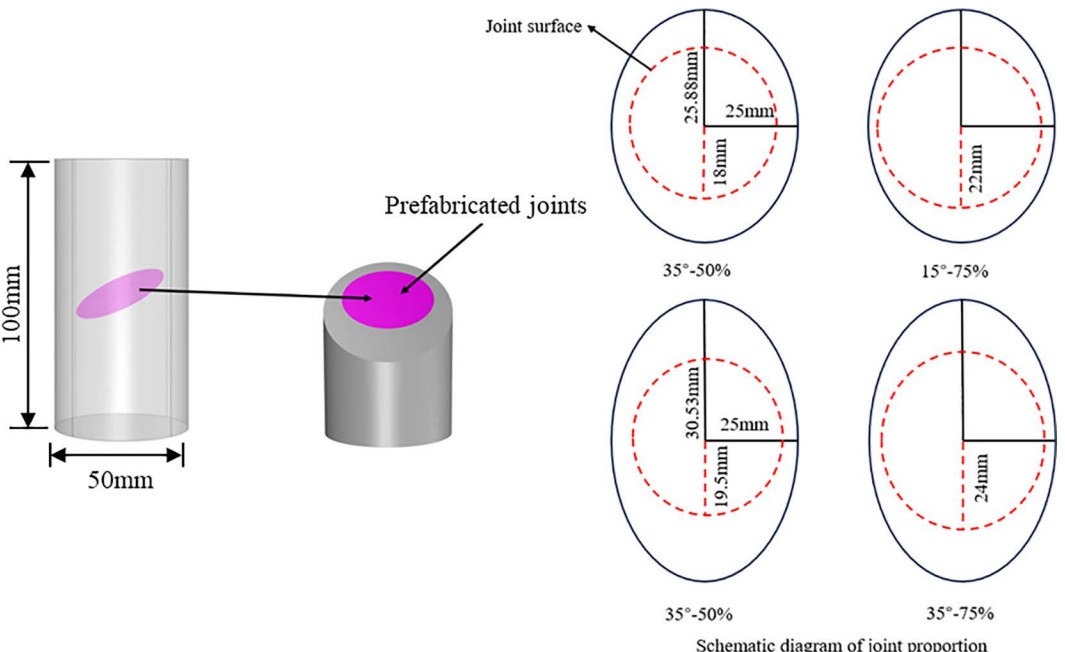

**Fig 4. Schematic diagram of prefabricated joints.**

a constant value but rather a function related to the wing crack propagation angle $\theta$, as well as the ratio of effective tangential stress to effective normal stress [38,39]. Introducing this function into the constitutive equation would complicate the equation and reduce its applicability. Furthermore, analysis of the formula reveals that it has no physical connection with the crack and

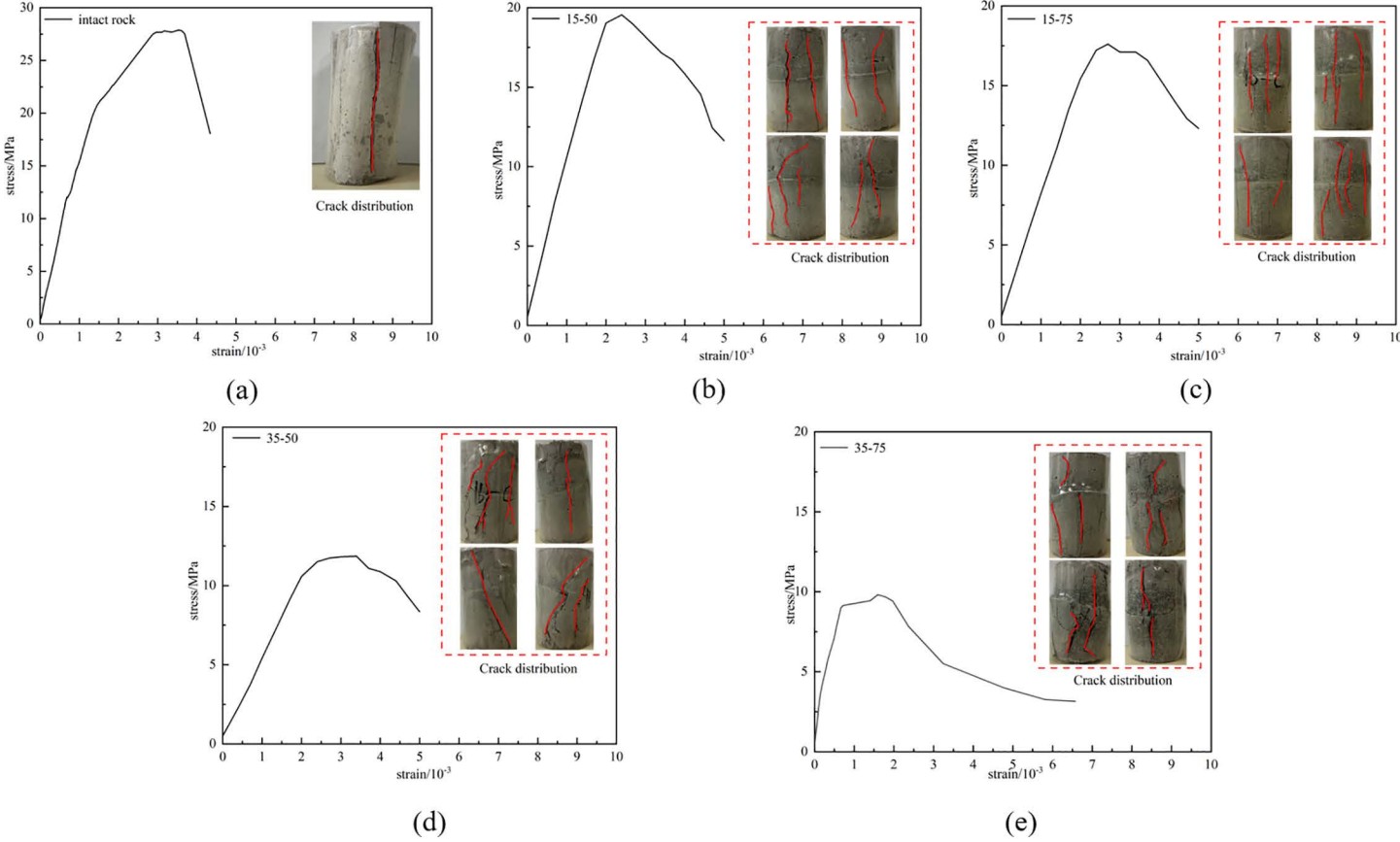

**Fig 5. Stress-strain diagram of clay-like rock.** (a) Intact rock, (b) 15°—50%, (c) 15°—75%, (d) 35°—50% and (e) 35°—75%.

can be treated as an adjustable parameter to better align with current experiments [35]. Therefore, this study solves for the correction coefficient based on the relationship between $\lambda$, the macroscopic damage variable, and the peak stress, as illustrated in Fig 6 and Fig 7. Fig 6 demonstrates that the value of $\lambda$ significantly influences the calculated results of the macroscopic damage variable, with both excessively large and small values leading to distorted outcomes. The direct consequence of macroscopic joints causing damage to the rock is a decrease in strength. Thus, using the relationship between $\lambda$ and the peak stress shown in Fig 7, and referencing the peak strengths of different specimens, the optimal value of $\lambda$ is determined through interpolation and presented in Table 1. Based on a comprehensive analysis, $\lambda = 0.32$ is adopted in this paper.

Simplifying Equations 19 and 20, we can obtain the calculation formulas for the macroscopic damage variable with a single crack under two different methods:

$$\begin{cases} D'_{II} = 1 - \dfrac{1}{1+ \frac{19.44(1-v^2)a\cos^2\alpha(\sin\alpha-\tan\phi\cos\alpha)^2}{V}\int_0^A \sec(\frac{\pi a}{w})dA + \frac{38.88a\cos^2\alpha(\sin\alpha-\tan\phi\cos\alpha)^2}{V}\int_0^l \sec(\frac{\pi a}{w})dl} \\ D''_{II} = 1 - \dfrac{1}{1+ \frac{9.436(1-v^2)a\cos^2\alpha(\sin\alpha-\tan\phi\cos\alpha)^2}{V}\int_0^A \sec(\frac{\pi a}{w})dA} \end{cases} \tag{22}$$

### 3.3 Verification of constitutive model

**3.3.1 Micro-damage constitutive model.** Zhao et al. [40] assumed that the strength of rock micro-elements follows a Weibull distribution and derived an expression for the mesoscopic damage variable based on the Drucker-Prager (D-P) failure criterion.

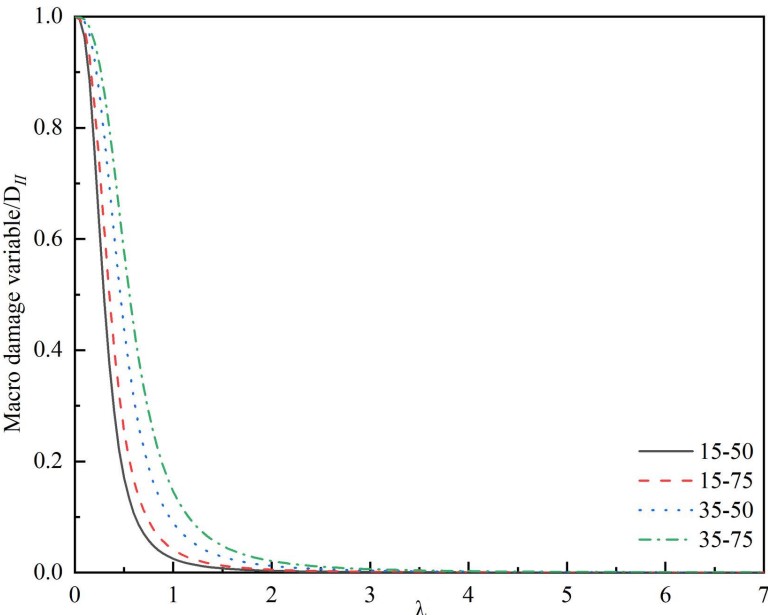

**Fig 6. Relationship between $\lambda$ and macro damage variable.**

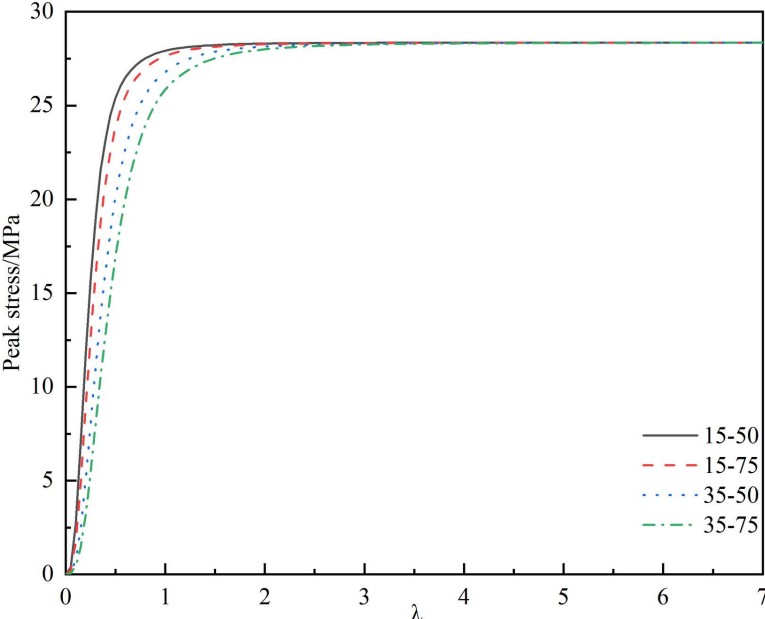

**Fig 7. Relationship between $\lambda$ and peak stress.**

**Table 1. Optimal λ under different conditions.**

|  | 15°—50% | 15°—75% | 35°—50% | 35°—75% | mean value |
|---|---|---|---|---|---|
| λ | 0.31 | 0.33 | 0.31 | 0.33 | 0.32 |

Expression for the damage variable:

$$D_I = 1 - \exp\left\{-\left(\frac{\left[\frac{(\sigma_1-\sigma_3)E_0\varepsilon_1}{\sqrt{3}(\sigma_1-2\nu\sigma_3)} + u\frac{(\sigma_1+2\sigma_3)E_0\varepsilon_1}{\sigma_1-2\nu\sigma_3} - \frac{\sqrt{3}C_0\cos\varphi}{\sqrt{3+\sin^2\varphi}}\right]}{F_0}\right)^m\right\}$$

(23)

where: $\varphi$ is the internal friction angle of the intact rock, $C_0$ is the cohesive force within the intact rock, $F_0$ and $m$ are the distribution parameters of Weibull function, $u$ is the material strength parameter, $u = \frac{\sin\varphi}{\sqrt{3}\sqrt{(3+\sin^2\varphi)}}$.

When performing a uniaxial compression test, the confining pressure $\sigma^3 = 0$. the statistical damage constitutive model for uniaxial compression tests is as follows:

$$\sigma_1 = E_0\varepsilon_1 \exp\left\{-\left(\frac{\left[\frac{E_0\varepsilon_1}{\sqrt{3}} + uE_0\varepsilon_1 - \frac{\sqrt{3}C_0\cos\varphi}{\sqrt{3+\sin^2\varphi}}\right]}{F_0}\right)^m\right\}$$

(24)

To verify the accuracy of the damage constitutive model, the experimental data of intact clayey rock from Section 3.1 is used for validation. The model parameters are listed in Table 2.

As can be seen from Fig 8, the experimental curve shows a high degree of fitting with the theoretical curve, indicating that the statistical damage constitutive model established based on the Drucker-Prager criterion can effectively reflect the stress-strain relationship during the loading process of intact rock.

**3.3.2 Macro-meso coupled damage constitutive model.** Fig 9 presents the stress-strain curves obtained using two calculation methods, which are derived by combining Equations 19, 20, 22, and 23. The calculation parameters are shown in Table 3. The results indicate that although there are still some deviations between the improved constitutive model curve and the experimental curve, the overall trend and peak stress of the curve are in good agreement with the experimental results, and are significantly better than before the improvement. By calculating the determination coefficients between the curves obtained by two different methods and the experimental curve, it can be intuitively seen that the improved constitutive equation has more advantages in describing the failure of internal joints, with an average determination coefficient of about 0.7, which fits well with the experimental curve. However, the predicted curve obtained from reference [31] and the experimental curve show that $R^2$ tends to zero, indicating that the model has poor predictability and cannot predict samples containing internal cracks well. In comparison, the improved constitutive model is more suitable for internal cracks.

**Table 2. The parameters of constitutive model.**

| $F_0$ /(MPa) | $m$ | $E_0$ /(GPa) | $v$ | $\varphi$ /(°) | $C_0$ /(MPa) |
|---|---|---|---|---|---|
| 39.835 | 1.4335 | 15.82 | 0.21 | 23 | 9.424 |

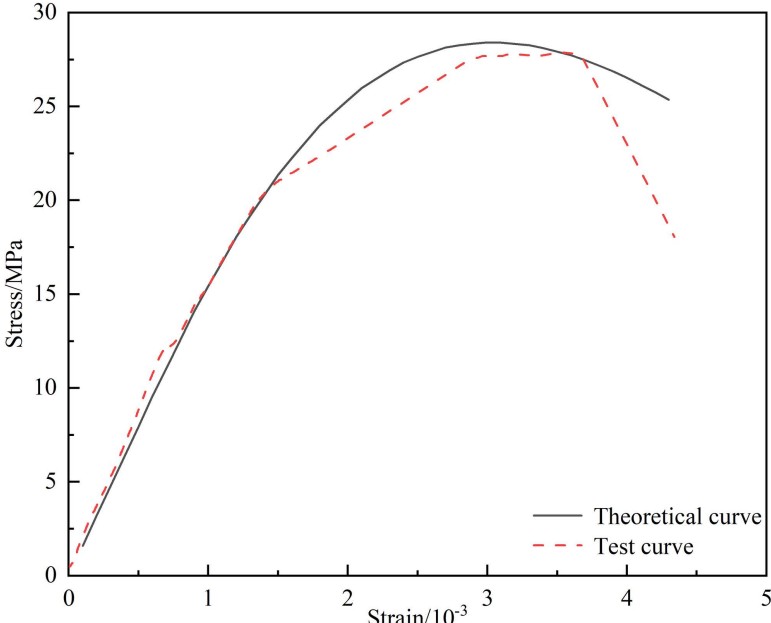

**Fig 8. Comparison diagram of experimental curve and theoretical curve.**

It is found that the experimental data from the 35°-75% test deviates significantly from the theoretical results. The stress-strain curves of 35°-75% specimens are found to contain a significant anomalous segment characterized by substantial strain variation with minimal stress increment, as illustrated in box a of Fig 9(d). Combined with further examination of the specimens, distinct shear sliding along the joint plane is identified in box b of Fig 9(d). For the phenomenon of abnormal data, the strength difference of the samples is caused by manual pouring, differences in mix proportions during the pouring process, and uneven mixing and vibration time. For subsequent experiments, we will strictly ensure the proportioning, control errors during the pouring process, and ensure consistent curing conditions to ensure the stability of sample strength.

From Fig 10, it can be observed that the damage variables calculated using the method from reference [31] are all smaller than those calculated using the improved method. A comparative analysis reveals that the peak stresses obtained using the improved method are more closely aligned with the experimental peak stresses, whereas those calculated using the method from reference [31] are generally larger. The reason for this phenomenon is that the stress conditions for internal cracks are more complex compared to those for penetrating cracks, which results in smaller damage variables.

In Fig 11, as strain gradually accumulates, the damage evolution trajectories during the loading process are exhibited to gradually converge and even almost overlap for both cracked rock samples and initially intact rock samples. Upon deeper analysis of this phenomenon, it is found that intact rocks undergo the initiation and propagation of macroscopic cracks after their peak strength is reached. This indicates that macroscopic and microscopic defect structures are present within the rocks after their peak strength is attained. Consequently, after peak strength, the differences in stress states between rocks that were initially intact and those with initial macroscopic cracks gradually diminish during continued loading, ultimately converging to similar residual strength levels.

Fig 12 illustrates the influence of crack inclination and crack length on the macroscopic damage variable and peak stress. It can be seen that both crack length and crack inclination affect the integrity of the rock. The peak stress exhibits a U-shaped variation with changes in crack inclination, while the macroscopic damage variable shows an inverted U-shaped

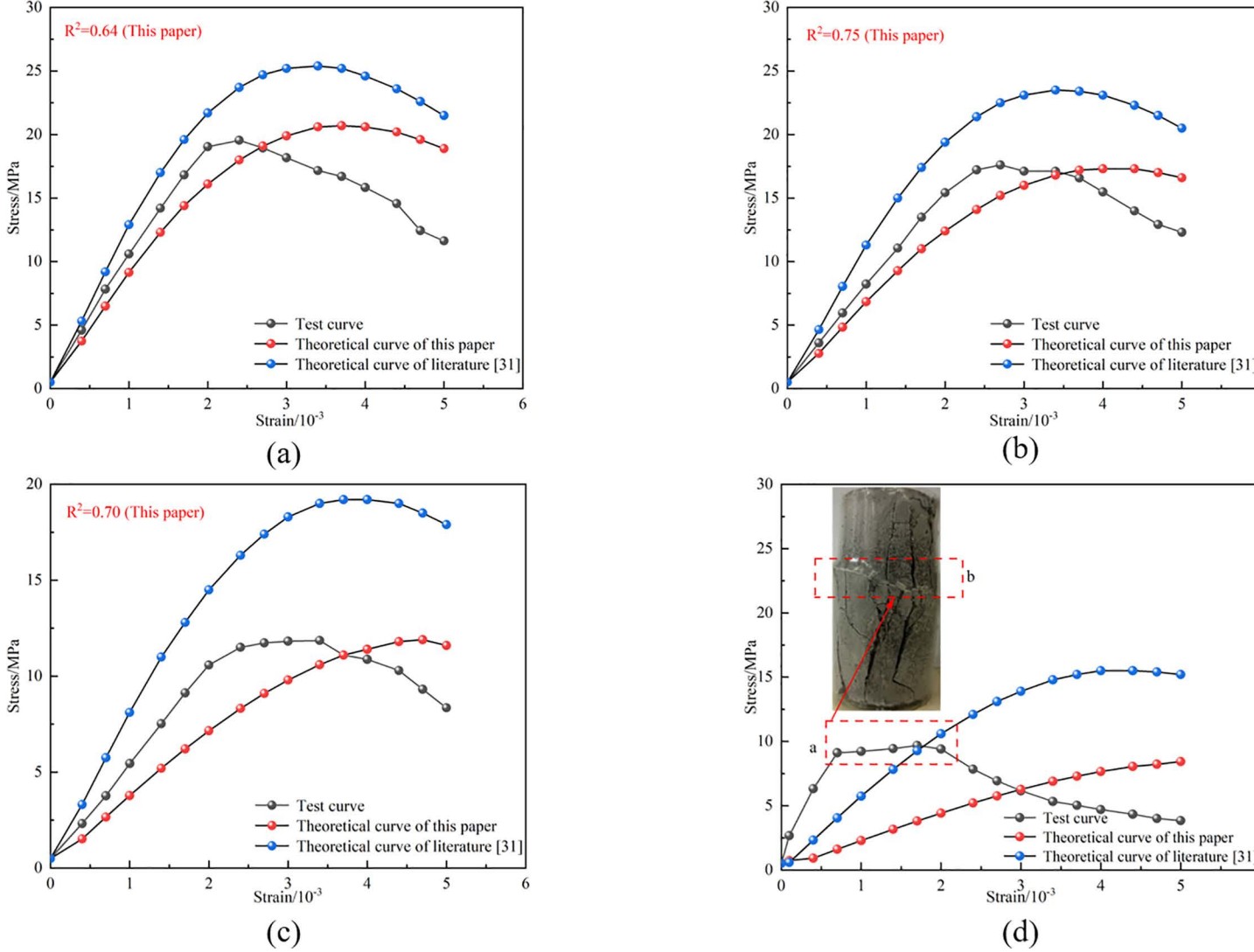

**Fig 9. Stress-strain diagram considering macro-mesoscopic damage constitutive model.** (a) 15°—50%, (b) 15°—75%, (c) 35°—50%and (d) 35°—75%.

**Table 3. Model calculation parameters.**

|  | $\alpha$ /(°) | $a$/ (cm) | $E_0$/ (GPa) | $v$ | $\phi$ /(°) | $W$ /(cm) | $V$ /(cm³) |
|---|---|---|---|---|---|---|---|
| 15°—50% | 15 | 1.8 | 15.82 | 0.21 | 0 | 5 | 50 |
| 15°—75% | 15 | 2.2 |  |  |  |  |  |
| 35°—50% | 35 | 1.95 |  |  |  |  |  |
| 35°—75% | 35 | 2.4 |  |  |  |  |  |

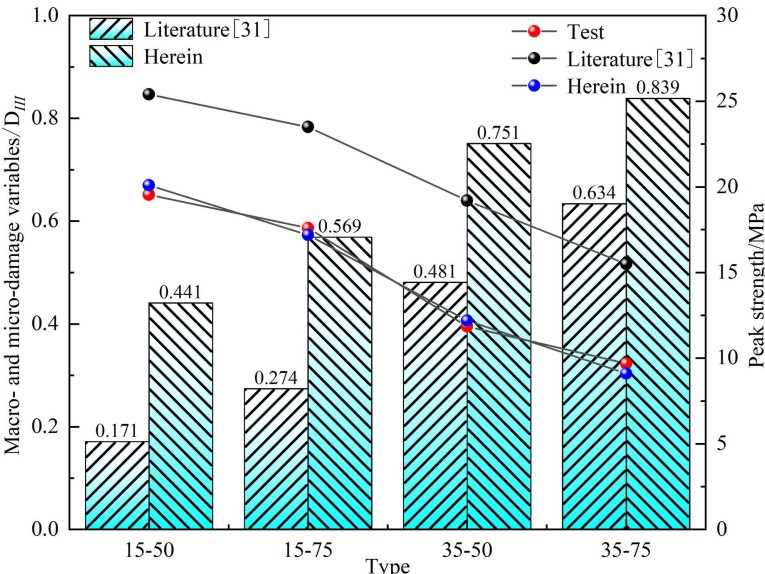

**Fig 10. Comparative Analysis of Macro-meso Coupling Damage Variables Based on Different Methods.**

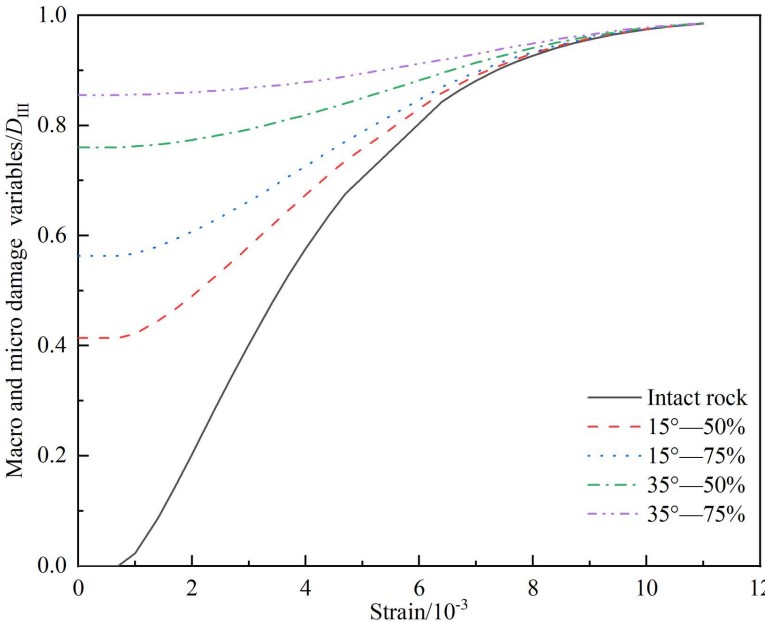

**Fig 11. Relationship curve of** $\varepsilon - D_{III}$

variation. The macroscopic damage variable reaches its maximum value at a crack inclination of 45°, where the peak stress is the lowest, and the differences in macroscopic damage variables before and after 45° are relatively small.

**3.3.3 Parameter sensitivity verification.** The investigation into the effect of different values of $\lambda$ on stress – strain, taking the 15°-75% specimen as an example, yields the results shown in the Fig 13. It is evident that different values of $\lambda$ exert a significant impact on the stress – strain curve. As $\lambda$ increases, the peak strength gradually rises and the

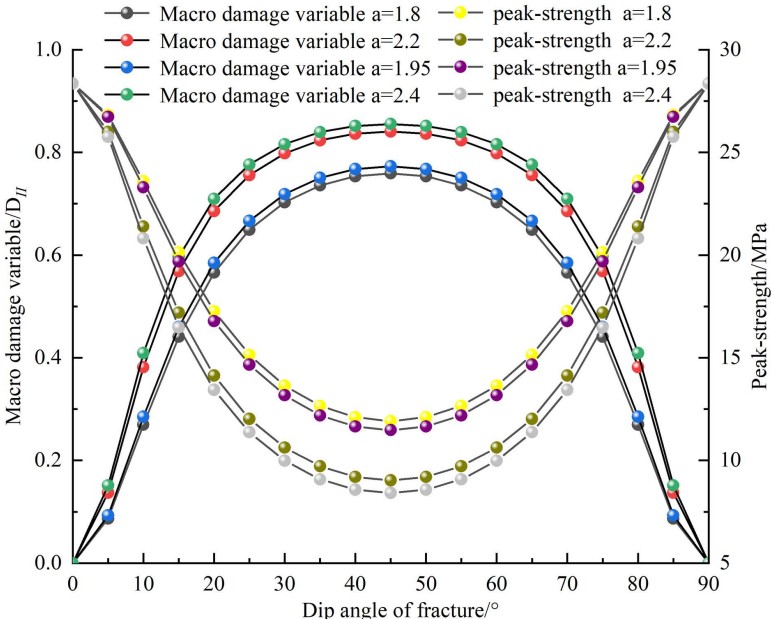

**Fig 12. Variations of macroscopic damage variable and peak strength of jointed rock mass with joint dip angle.**

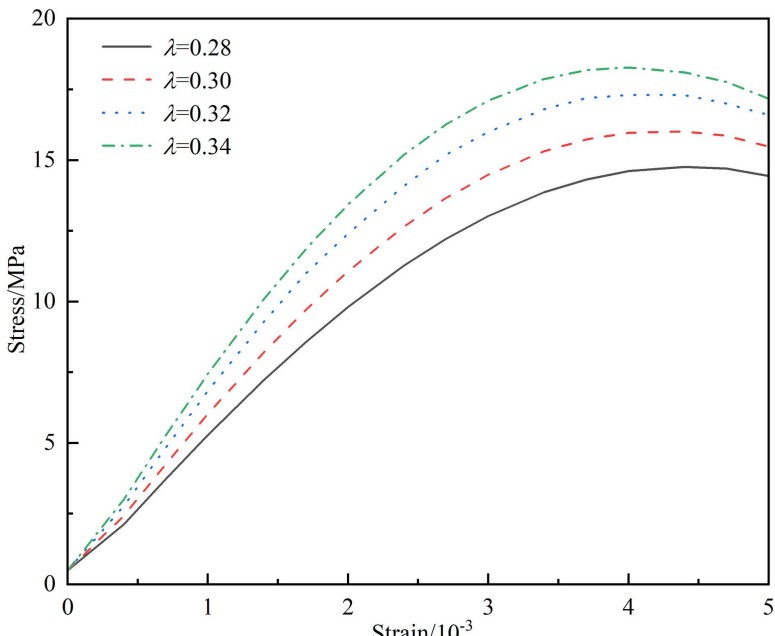

**Fig 13. The influence of different $\lambda$ values on the stress-strain curve.**

corresponding peak strain gradually decreases. This indicates that under identical conditions, the strength of the rock gradually increases and its brittleness becomes more pronounced. This is in line with the previous analysis of the influence of $\lambda$ on the formation of prefabricated joints. Increasing $\lambda$ reduces the damage caused by the formation of prefabricated joints and enhances the strength of the entire specimen.

The influence of the characteristic parameter *m* in the Weibull function on the macroscopic and microscopic coupled damage variables, as well as the stress-strain curve, is illustrated by Fig 14 and Fig 15, taking the 15°-50% sample as an example. It is revealed by Fig 14 that under different *m* values, a critical point exists where the coupled macroscopic and microscopic damage variables become equal. Before this critical point, the coupled damage variable is gradually decreased as *m* increases; however, beyond the critical point, a larger m results in a greater coupled damage variable. Furthermore, the impact of *m* on the stress-strain curve is demonstrated by Fig 15: as *m* increases, the peak stress also rises. Additionally, the stress-strain curve tends to flatten with increasing *m* prior to reaching the peak stress, while it becomes steeper after the peak, indicating that the brittleness and ductility of the rock are significantly affected by *m*. The concentration degree of the rock's micro-element strengths is characterized by *m* [41].

The influence of the characteristic parameter $F_0$ in the Weibull function on the macroscopic and microscopic coupled damage variables, as well as the stress-strain curve, is illustrated in Fig 16 and Fig 17, with the 15°-50% sample serving as an example. It is shown by Fig 16 that the coupled damage variable gradually decreases as $F_0$ increases. Fig 17 indicates that with the increment of $F_0$, both the peak stress and its corresponding peak strain are gradually increased. The pre-peak portion of the stress-strain curve is minimally impacted by the increase in $F_0$, suggesting that $F_0$ reflects the average magnitude of the macroscopically statistically determined micro-element strengths in the rock [41].

Taking 35°—50% as an example, the influence of different friction angles within the joint plane on the stress-strain curve was explored, and the results are shown in Fig 18 From the figure, it can be seen that as the friction angle within the joint surface increases, the strength of the specimen also increases This is because the internal friction angle within the joint surface suppresses joint slip, thereby improving the strength of the specimen At the same time, when the internal friction angle of the joint surface approaches the joint dip angle, the influence of the joint on the specimen becomes very small, and the stress-strain curve of the specimen changes with a high degree of coincidence with the intact specimen This is because when the internal friction angle of the joint surface is greater than or close to the joint dip angle, the shear force on the joint is greater than or close to the anti slip force provided by the joint surface, so the influence of the joint on the specimen will be very small

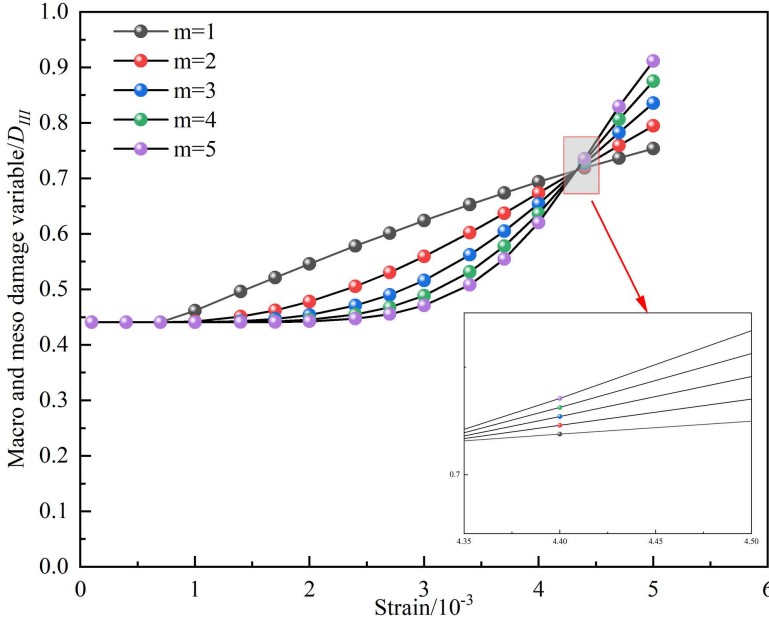

**Fig 14. Effect of *m* on coupling damage factor.**

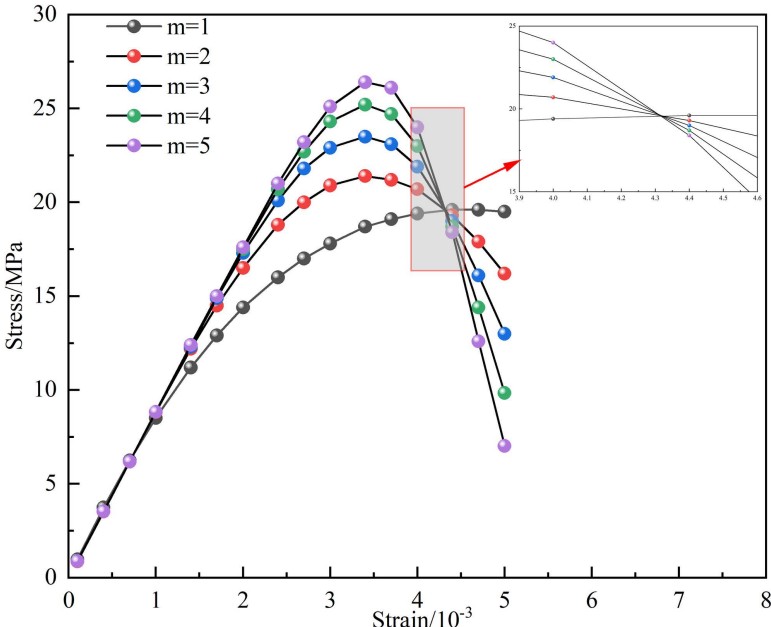

**Fig 15. Effect of *m* on stress-strain curve.**

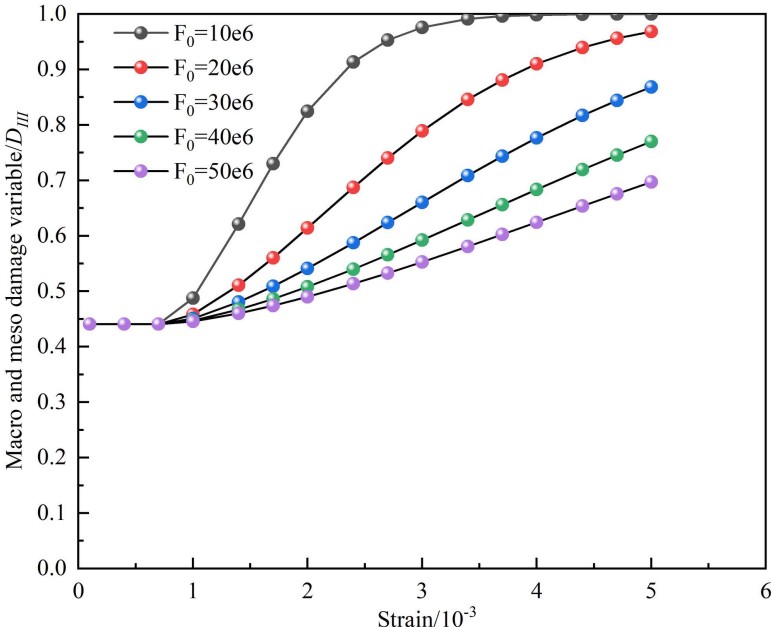

**Fig 16. Effect of *F0* on coupling damage factor.**

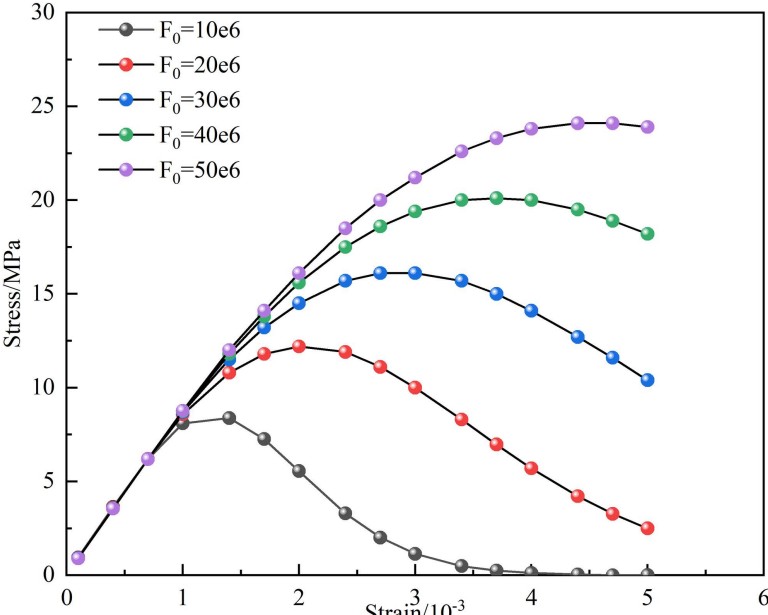

**Fig 17. Effect of *F0* on stress-strain curve.**

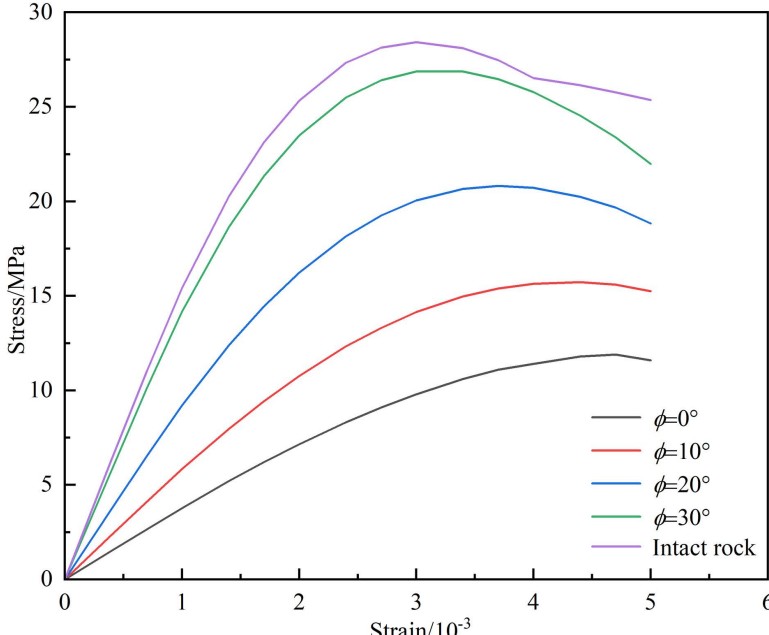

**Fig 18. The influence of friction angle $\phi$ within the joint surface on stress-strain.**

## 4. Discussion

With the rapid expansion of urban scale and the increasing scarcity of land resources, the development and utilization of underground space have emerged as a critical means to alleviate land resource constraints [42] However, an inevitable challenge in underground space development is the stability of surrounding rock masses, which directly impacts the safety of underground engineering projects In nature, rocks inherently exhibit micro-voids and other microscopic damages, along with macroscopic defects such as joints The damage constitutive equation established in this study incorporates both macroscopic and mesoscopic damages, offering significant implications for evaluating the stability of underground engineering, predicting the strength and deformation characteristics of surrounding rock masses in underground structures Nevertheless, to facilitate the solution of the constitutive model, certain idealized assumptions were made, such as assuming that the final propagation direction of wing cracks is parallel to the applied axial force direction and idealizing the internal friction angle of joint surfaces These assumptions may deviate from actual conditions, thereby affecting the model's applicability

The present model demonstrates good applicability under conventional conditions, including quasi-static loading, room temperature, and uniaxial stress states However, for scenarios involving dynamic loading, temperature variations, or multiaxial stress conditions, the model necessitates the incorporation of strain-rate-dependent terms or thermomechanical coupling effects [43–45] For multi axis states, the anisotropy of joints in three-dimensional space needs to be considered, such as introducing the concept of tensors to calculate and characterize the degree of joint damage [46,47] Additionally, there exists a scale discrepancy between laboratory specimens and field applications, as laboratory samples often feature small dimensions and idealized boundaries, contrasting with the heterogeneous nature and complex loading conditions of actual structures To bridge this gap, similarity theory, multiscale modeling, or in-situ monitoring are required to validate the model's applicability Furthermore, environmental factors such as the heterogeneity of geological conditions, groundwater presence, and temperature fluctuations, as well as dynamic loading conditions, may necessitate the introduction of corrective terms or segmented modeling approaches into the model to enhance prediction accuracy

## 5. Conclusion

1) An improved macroscopic and microscopic damage constitutive model has been proposed, which comprehensively considers the geometric and mechanical parameter characteristics of cracks Assuming that the strength of rock microelements follows the Weibull distribution, based on the Drucker Prager (D-P) strength criterion, the concept of evolutionary fracture is introduced, and the macroscopic microscopic coupling effect is considered to establish a damage constitutive model for rock masses with internal fractures

2) The new stress intensity factor is introduced to characterize the tips of internal cracks, and the model incorporates the consideration of stress-induced crack propagation It effectively predicts the peak strength and the stress-strain relationship of rocks When compared to existing methods for calculating damage variables, this approach offers a superior explanation for the initial damage resulting from internal cracks

3) The in-depth analysis was conducted on the parameters involved in the constitutive model, with a focus on exploring the impact of these model parameters on the model's behavior and predictive ability Here, $m$ reflects the concentration of rock micro-element strength, while $F_0$ reflects the average strength of rock micro-elements In addition, parameters $\phi$ can affect the properties of the joint surface, thereby affecting the strength of the specimen

| Nomenclature | |
|---|---|
| **Latin Symbols** | |
| $D_I$ | Meso damage of intact rock caused by loading |
| $D_{II}$ | Damage caused by prefabricated joints |
| $D_{III}$ | Macro-meso coupling damage variable |
| $U$ | Rrelease rate of damage strain energy |

**Nomenclature**

| | |
|---|---|
| $E$ | Elastic modulus |
| $U^E$ | Elastic strain energy per unit volume |
| $\Delta U^E$ | Change in elastic strain energy per unit volume |
| $\Delta U_1$ | strain energy caused by cracking of joints and fissures |
| $\Delta U_2$ | strain energy caused by further expansion after joint cracking and sliding |
| $V$ | Volume of the loading test |
| $K_I$ | Mode I intensity factor of wing crack at the crack tip |
| $K_{II}$ | Mode II intensity factor of wing crack at the crack tip |
| $A$ | Surface area of the crack |
| $v$ | Poisson's ratio |
| $G$ | Crack surface release rate |
| $W$ | Flat width |
| $E_0$ | Elastic modulus of intact rock |
| $a$ | Half length of the crack |
| $l$ | Extension length of wing crack |
| $C_0$ | Cohesive force within the intact rock |
| $F_0, m$ | Distribution parameters of Weibull function |
| $u$ | Material strength parameter |

**Greek symbols**

| | |
|---|---|
| $\sigma_1$ | Axial stress |
| $\varepsilon_1$ | Axial strain |
| $\sigma_3$ | confining pressure |
| $\sigma_3^i$ | Internal stress |
| $\sigma_n$ | Normal stress |
| $\tau_n$ | Shear stress |
| $\tau_f$ | Joint slip shear stress |
| $\lambda$ | Correction factor |
| $\phi$ | Friction angle of the structural plane |
| $\mu$ | Coefficient of friction |
| $\alpha$ | Dip angle of fracture |
| $\theta$ | Starting angle of wing crack |
| $\varphi$ | Internal friction angle of the intact rock |

**Abbreviations**

| | |
|---|---|
| 15°—50% | Joint dip angle of 15 degrees, joint proportion of 50% |
| 15°—75% | Joint dip angle of 15 degrees, joint proportion of 75% |
| 35°—50% | Joint dip angle of 35 degrees, joint proportion of 50% |
| 35°—75% | Joint dip angle of 35 degrees, joint proportion of 75% |

## Author contributions

**Conceptualization:** Miao He.

**Formal analysis:** Hongliang Zhao, Yiru Zhang.

**Investigation:** Zhiying Gong, Hao Guan.

**Methodology:** Qingrui Lu, Shuren Hao.

**Writing – original draft:** Haian Liang, Miao He.

**Writing – review & editing:** Haian Liang.

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
