## [Decision Letter · Decision Letter 0]

PONE-D-25-16468A Macro-meso Damage Coupling Rock Mass Damage Model Based on Improved Internal Crack AnalysisPLOS ONE

Dear Dr. He,

Thank you for submitting your manuscript to PLOS ONE. After careful consideration, we feel that it has merit but does not fully meet PLOS ONE’s publication criteria as it currently stands. Therefore, we invite you to submit a revised version of the manuscript that addresses the points raised during the review process.

We look forward to receiving your revised manuscript.

Kind regards,

Kang Wang, Ph.D.

Academic Editor

PLOS ONE

Journal Requirements:

the National Natural Science Foundation of China (Grant No. 42077255)�the National Natural Science Foundation of China (Grant No. 42002258)

3. We note that your Data Availability Statement is currently as follows: within the manuscript and/or Supporting Information files

Reviewers' comments:

Reviewer's Responses to Questions

**Comments to the Author**

1. Is the manuscript technically sound, and do the data support the conclusions?

Reviewer #1: Yes

Reviewer #2: Yes

Reviewer #3: Yes

2. Has the statistical analysis been performed appropriately and rigorously? 

Reviewer #1: Yes

Reviewer #2: Yes

Reviewer #3: I Don't Know

3. Have the authors made all data underlying the findings in their manuscript fully available?

Reviewer #1: Yes

Reviewer #2: Yes

Reviewer #3: Yes

4. Is the manuscript presented in an intelligible fashion and written in standard English?

Reviewer #1: No

Reviewer #2: Yes

Reviewer #3: Yes

5. Review Comments to the Author

Reviewer #1: Comments

The paper entitled “A Macro-meso Damage Coupling Rock Mass Damage Model Based on Improved Internal Crack Analysis” presents a macro-meso damage coupling rock mass damage model based on improved internal crack analysis. The manuscript appears to make a valuable contribution to rock mechanics and underground engineering by developing a damage constitutive model that bridges macro and micro scales. However, there are some major issues for consideration. Please see my comments below:

1. The introduction lacks sufficient clarity in identifying the specific research gap this study addresses within the well-established field of damage modeling. Statistical damage models have been extensively developed over decades, making it crucial to articulate the novel contributions of this work. The manuscript should: clearly delineate how this approach differs from existing models in the literature, address and acknowledge relevant pioneering work, such as Hongyan Liu's contributions from China University of Geosciences, provide a more comprehensive literature review that positions this research within the broader scientific context, explicitly state the limitations of previous models that this study aims to overcome.

2. The current bibliography demonstrates a significant imbalance with an overrepresentation of Chinese references. Incorporate more diverse international references reflecting global advancements in rock mechanics and damage modeling. For example, A plastic strain based statistical damage model for brittle to ductile behaviour of rocks.

3. Many typos have been found in the current version, please double-check and correct them. For example, “Building upon fracture and damage theories”, “the Surface additional strain energy”.

4. In Equation (1), many previous studies have mentioned this equation, please add reletated references.

5. The validation study using prefabricated joint specimens lacks essential methodological details. The authors should provide precise specimen dimensions, explain the technical procedure for preparing internal fractures (acknowledging this as a challenging aspect), and describe how fracture mechanical properties were determined. These details are crucial for ensuring experimental reproducibility and evaluating the validity of the results.

6. The verification approach would be significantly strengthened by including comparative analysis between the proposed model and existing established models. The authors should quantify improvements in predictive accuracy using appropriate statistical measures and provide visual comparisons showing how their model performs against others under identical conditions. This comparative approach would more convincingly demonstrate any advantages of the proposed methodology.

7. Finally, the manuscript should include a dedicated section addressing the limitations of the study. The authors should acknowledge specific boundary conditions under which the model is valid, discuss scaling issues between laboratory specimens and field applications, and identify geological or loading conditions where the model might require modifications.

Reviewer #2: The manuscript proposes a macro-meso damage constitutive model that integrates improved internal crack analysis with statistical damage theory and experimental validation. The study is relevant to rock mechanics and underground engineering, especially for predicting failure in jointed rock masses.

1、 Clarity and Organization of Derivations:

While the derivations are comprehensive, some sections (particularly Equations 2–10) might benefit from additional verbal explanation. More explicit discussion of the assumptions underlying each derivation would help readers who are not experts in damage mechanics follow the logical progression.

2、 Notation and Consistency:

The manuscript uses a variety of symbols and subscripts (e.g., DI, DII, DIII) that are critical to the model formulation. A consolidated table of symbols and parameters might enhance readability and ensure consistency throughout the paper.

3、 Complexity in the Stress Intensity Factor Reformulation:

The introduction of a new stress intensity factor to address the singularity problem is an important innovation. However, further discussion about the physical implications of introducing the adjustable parameter λ and its variation with geometric parameters would be useful. Clarifying the limitations of assuming a constant or nearly constant λ in different loading conditions might improve the overall understanding.

4、 Discussion on Model Applicability and Limitations:

While the manuscript convincingly shows that the improved model better fits the experimental data for the given clay-like rocks, it would be beneficial to address:

How the model might behave under different loading paths (e.g., triaxial tests) or different rock types.

Any limitations regarding scale effects or the assumption of an infinite matrix when deriving the stress intensity factors.

A more explicit discussion on the potential applicability to real field conditions and how uncertainty in parameter estimation might influence predictions.

5、 While the introduction explains the significance of the study, it could benefit from more in-depth references to recent research in the past five years. Highlight gaps in the literature this study addresses, such as: compgeo.2024.106095; tafmec.2024.104691; fuel.2023.129584

6、 Experimental Uncertainties:

The experiments are well described; however, the treatment of experimental uncertainties or data scatter—especially in the context of specimens that exhibit joint-induced premature failure—could be better addressed. Quantifying error margins or providing a brief statistical analysis would lend further credibility to the validation.

7、 Enhanced Explanations in the Theoretical Section:

Consider adding a flowchart or diagram summarizing the derivation steps from the initial damage formulation to the final constitutive equation.

Offer a brief commentary after each set of key equations to contextualize the physical meaning of the terms.

8、 Symbol and Parameter Table:

Include a dedicated table that lists all symbols, parameters, and their units. This would help readers quickly reference the notation used throughout the derivations.

9、 Discussion on the Correction Coefficient λ:

Expand on the reasoning behind the chosen value of λ = 0.32 and discuss the sensitivity of the model outputs to variations in λ. A brief parametric study or discussion on the impact of different λ values in various scenarios could be valuable.

10、 Extend the Experimental Discussion:

Elaborate on the observed discrepancies in the 35°–75% test data. Discuss potential causes such as specimen preparation variability, and suggest how future studies could control or account for such effects.

If available, include additional statistical analyses (e.g., standard deviation or confidence intervals) to accompany the experimental data.

Reviewer #3: 1. The current assumption that crack propagation direction is always parallel to the principal stress is oversimplified. It is recommended to validate the universality of this hypothesis through multiple comparative experiments.

2. The current model is limited to uniaxial compression conditions. It is suggested to supplement validation experiments under multi-axial stress conditions.

3. Some curves in the figures lack clarity in annotations. It is recommended to add local enlarged views or arrow labels for abnormal regions.

4. Adjust the placement of certain figures to follow their corresponding textual descriptions.

5. The conclusions require stricter phrasing to avoid overgeneralized statements.

6. It is recommended to enhance the discussion on practical engineering applicability.

7. Revise grammatical errors throughout the manuscript and unify the reference format.

8. Supplement citations of critical literature to highlight the novelty and improvements of this work.

6. PLOS authors have the option to publish the peer review history of their article (what does this mean? ). If published, this will include your full peer review and any attached files.

**Do you want your identity to be public for this peer review?** For information about this choice, including consent withdrawal, please see our Privacy Policy .

Reviewer #1: No

Reviewer #2: No

Reviewer #3: No

---

## [Author Response · Author response to Decision Letter 1]

15 May 2025

Reviewer #1

Comment:

Point 1: The introduction lacks sufficient clarity in identifying the specific research gap this study addresses within the well-established field of damage modeling. Statistical damage models have been extensively developed over decades, making it crucial to articulate the novel contributions of this work. The manuscript should: clearly delineate how this approach differs from existing models in the literature, address and acknowledge relevant pioneering work, such as Hongyan Liu's contributions from China University of Geosciences, provide a more comprehensive literature review that positions this research within the broader scientific context, explicitly state the limitations of previous models that this study aims to overcome.

Response: Many thanks for your advice.

We have added an overall overview of the current research field, including the main research directions and achievements in this area. A detailed list of key literature directly related to this study was provided, along with a brief overview of their main viewpoints and findings, in order to provide readers with a clear research background. At the same time, the introduction has been restructured to ensure clear logic and natural transitions between paragraphs. Redundant information that is not directly related to the topic has been removed, making the introduction more concise and focused on the core research question. (Page 3-4, lines 29-60 of the revised manuscript). Added references [12,13,17,18,19,22,23,24] in the revised manuscript.

Point 2: The current bibliography demonstrates a significant imbalance with an overrepresentation of Chinese references. Incorporate more diverse international references reflecting global advancements in rock mechanics and damage modeling. For example, A plastic strain based statistical damage model for brittle to ductile behaviour of rocks.

Response: Many thanks for your advice.

We recognize that in academic research, extensive citation of foreign literature can more comprehensively showcase the cutting-edge trends and advanced research methods in this field internationally. Previously, the proportion of Chinese literature in the paper was relatively high, which may have limited the international perspective of the paper to some extent. Therefore, we actively take measures to increase the citation of foreign literature in order to enhance the academic depth and breadth of our paper. Added references [12,13,17,18,19,24,27,37,38,39,40,41,42] in the revised manuscript.

Point 3: Many typos have been found in the current version, please double-check and correct them. For example, “Building upon fracture and damage theories”, “the Surface additional strain energy”.

Response: Many thanks for your advice.

Incorrect spelling not only affects the readability of the paper, but may also raise doubts among readers about the professionalism and rigor of the paper, thereby affecting the dissemination and recognition of research results. We have carefully read the entire text and corrected any errors in wording and wording.

Point 4: In Equation (1), many previous studies have mentioned this equation, please add reletated references.

Response: Many thanks for your advice.

We have added a reference to formula (1). (Reflected in reference [27] of the revised manuscript)

Point 5: The validation study using prefabricated joint specimens lacks essential methodological details. The authors should provide precise specimen dimensions, explain the technical procedure for preparing internal fractures (acknowledging this as a challenging aspect), and describe how fracture mechanical properties were determined. These details are crucial for ensuring experimental reproducibility and evaluating the validity of the results.

Response: Many thanks for your advice.

We have added a schematic diagram of the sample pouring process in the text (Page 11, In Figure 3 of the revised version) and labeled the sample size information (Page 12, In Figure 4 of the revised version) to provide readers with a clear understanding of the pouring process.

For the determination of the mechanical properties of joints, according to the final expression obtained, the only parameter involved in joints is the internal friction angle � of the joint. Since prefabricated internal joints are formed by filling composite paper and the joints are in an open state without contact at both ends(The sample preparation process is shown in Page 11, In Figure 3 of the revised version), this article takes the internal friction angle of the joint surface as 0°. However, it can be seen that the internal friction angle of the joint surface has an impact on the entire constitutive model, so discussions on different internal friction angles are added. (Page 22-23, lines 304-313 of the revised manuscript)

Point 6: The verification approach would be significantly strengthened by including comparative analysis between the proposed model and existing established models. The authors should quantify improvements in predictive accuracy using appropriate statistical measures and provide visual comparisons showing how their model performs against others under identical conditions. This comparative approach would more convincingly demonstrate any advantages of the proposed methodology.

Response: Many thanks for your advice.

In order to compare the model established in the article and make it more suitable compared to before the improvement, the coefficient of determination R ^ 2 was used for quantitative characterization, highlighting that the improved model is more suitable for the case of internal joints. (Page 17-18, lines 242-247 of the revised manuscript).

Point 7: The manuscript should include a dedicated section addressing the limitations of the study. The authors should acknowledge specific boundary conditions under which the model is valid, discuss scaling issues between laboratory specimens and field applications, and identify geological or loading conditions where the model might require modifications.

Response: Many thanks for your advice.

A discussion section has been added to the article, which analyzes and discusses the limitations of the current research, as well as other issues such as the differences between indoor experiments and on-site discussions. (Page 23-24, lines 314-343 of the revised manuscript).

Reviewer #2

Comment:

Point 1: While the derivations are comprehensive, some sections (particularly Equations 2–10) might benefit from additional verbal explanation. More explicit discussion of the assumptions underlying each derivation would help readers who are not experts in damage mechanics follow the logical progression.

Response: Many thanks for your advice.

A more detailed interpretation of the derivation was provided for ease of reading, with a focus on explaining formula (10).

Point 2: The manuscript uses a variety of symbols and subscripts (e.g., DI, DII, DIII) that are critical to the model formulation. A consolidated table of symbols and parameters might enhance readability and ensure consistency throughout the paper.

Response: Many thanks for your advice.

A symbol table has been added to the article, which includes the symbols used in the text. (Page 1-2 of the revised manuscript).

Point 3: The introduction of a new stress intensity factor to address the singularity problem is an important innovation. However, further discussion about the physical implications of introducing the adjustable parameter λ and its variation with geometric parameters would be useful. Clarifying the limitations of assuming a constant or nearly constant λ in different loading conditions might improve the overall understanding.

Response: Many thanks for your advice.

The introduction of adjustable parameters into the constitutive equation endows the model with flexibility in describing multi condition specimens, enabling it to adapt to different damage states through parameter fitting, improve the prediction accuracy of damage accumulation, failure evolution, and macroscopic mechanical response, and simplify complex problems through experience or equivalent parameters. Although some parameters may lack direct physical meaning, parameter sensitivity analysis can reveal their potential physical essence (Page 20, lines 278-285 of the revised manuscript).and provide a basis for material design and optimization, thus balancing engineering practicality and scientific interpretability.

Assuming that the adjustable parameters in the constitutive equation are constant or close to constant under complex load conditions simplifies the model, but it may result in the model being unable to describe the rate dependence in dynamic loading, damage accumulation and path dependence under cyclic loading, phase transition and performance degradation caused by temperature changes, effects under multiaxial stress, and creep under long-term loading, thereby reducing prediction accuracy and potentially misleading engineering safety assessments.

Point 4: While the manuscript convincingly shows that the improved model better fits the experimental data for the given clay-like rocks, it would be beneficial to address:

1、 How the model might behave under different loading paths (e.g., triaxial tests) or different rock types.

Response: Many thanks for your advice.

This article is derived based on the problem of plane stress, which is a two-dimensional state. To consider the triaxial compression state, the jointed specimen needs to be analyzed in a three-dimensional state, taking into account the anisotropy of the joints. Therefore, the damage constitutive model derived in this article is not applicable to three-dimensional states. It has been explained in the discussion section. (Page 23-24, lines 315-340 of the revised manuscript). The constitutive equation for damage in three-dimensional state has also been derived and some results have been obtained. However, due to the length limitation of the article, other articles will be used for derivation in the future to improve the entire system. According to existing research, the macroscopic damage caused by prefabricated cracks can be expressed as:

1

where�I is a fourth-order unit tensor�CL is the equivalent flexibility tensor of jointed rock mass�C0 is the flexibility tensor of the intact rock mass

The constitutive equation for linear elasticity is:

2

The fourth-order flexibility tensor can be expressed as:

3

Due to the symmetry characteristic of the stress tensor, i.e. �ij=�ji, it can be rewritten as: it can be rewritten as:

4

Similarly, continuing to apply the symmetry of the strain tensor (�ij=�ji) and the symmetry of the flexibility matrix (Cijkl=Cijlk=Cjikl=Cjilk) can further simplify:

5

By selecting an appropriate damage tensor expression to characterize the joints and substituting it into the equation, a three-dimensional damage constitutive equation containing macroscopic and microscopic damage variables can be obtained.

(a) 15°-50%(uniaxial compression) (b) 15°-75% (uniaxial compression)

(c) 15°-50%(Three axis compression -10MPa)

The above figure shows some of the research progress that has been made, and further analysis will be conducted in other articles in the future.

2、Any limitations regarding scale effects or the assumption of an infinite matrix when deriving the stress intensity factors.

The proportional effect and infinite matrix assumption simplify the analysis process in derivation, but their limitations (such as small-scale yielding, ignoring finite size effects, etc.) lead to a decrease in the accuracy of solutions under complex engineering conditions.

3、 A more explicit discussion on the potential applicability to real field conditions and how uncertainty in parameter estimation might influence predictions.

We have added a discussion section where we discussed the practical applicability of the model and the impact of uncertainty. (Page 23-24, lines 315-340 of the revised manuscript).

Point 5: While the introduction explains the significance of the study, it could benefit from more in-depth references to recent research in the past five years. Highlight gaps in the literature this study addresses, such as:compgeo.2024.106095; tafmec.2024.104691; fuel.2023.129584

Response: Many thanks for your advice.

We have added the research results of the past five years in the introduction section to highlight the supplement and further improvement of this study to existing research (Page 3-4, lines 29-60 of the revised manuscript).

Point 6: The experiments are well described; however, the treatment of experimental uncertainties or data scatter—especially in the context of specimens that exhibit joint-induced premature failure—could be better addressed. Quantifying error margins or providing a brief statistical analysis would lend further credibility to the validation.

Response: Many thanks for your advice.

We quantified the goodness of fit by calculating the coefficient of determination R ^ 2, making it more intuitive to see the improvement. (Page 17-18, lines 242-247 of the revised manuscript).

Point 7: Enhanced Explanations in the Theoretical Section:

1、 Consider adding a flowchart or diagram summarizing the derivation steps from the initial damage formulation to the final constitutive equation.

2、 Offer a brief commentary after each set of key equations to contextualize the physical meaning of the terms.

Response: Many thanks for your advice.

We have added a flowchart of the derivation in the article, which allows us to quickly understand the overall idea and method of the derivation. (Page 5, In Figure 1 of the revised version)

Point 8: Include a dedicated table that lists all symbols, parameters, and their units. This would help readers quickly reference the notation used throughout the derivations.

Response: Many thanks for your advice.

A symbol table has been added to the article, which includes the symbols used in the text. (Page 1-2 of the revised manuscript).

Point 9: Expand on the reasoning behind the chosen value of λ = 0.32 and discuss the sensitivity of the model outputs to variations in λ. A brief parametric study or discussion on the impact of different λ values in various scenarios could be valuable.

Response: Many thanks for your advice.

A more detailed analysis and reasoning were conducted on how to choose �, and the changes in the constitutive equation under different � conditions were added. The influence of � on the constitutive equation was discussed. (Page 20, lines 278-285 of the revised manuscript).

Point 10: Extend the Experimental Discussion:

1、 Elaborate on the observed discrepancies in the 35°–75% test data. Discuss potential causes such as specimen preparation variability, and suggest how future studies could control or account for such effects.

2、 If available, include additional statistical analyses (e.g., standard deviation or confidence intervals) to accompany the experimental data.

Response: Many thanks for your advice.

Added abnormal analysis of 35°-75% sample data in the middle, analyzed the reasons for the abnormalities, and summarized them, providing some feasible solutions for future experiments. (Page 18, lines 252-256 of the revised manuscript).

Reviewer #3

Comment:

Point 1: The current assumption that crack propagation direction is always parallel to the principal stress is oversimplified. It is recommended to validate the universality of this hypothesis through multiple comparative experiments.

Response: Many thanks for your advice.

I completely agree with the viewpoint you have put forward. Many scholars have conducted relevant research on the angle of ultimate crack propagation [1,2]. This article assumes that the ultimate crack propagation is parallel to the loading direction, which is a more ideal situation, and draws on the treatment method in literature [3]. The study of the angle of ultimate cracking and expansion of joints can provide a more accurate prediction of the constitutive model, and we will further improve this issue in future research. References have been added to the article (Page 7, lines 109 of the revised manuscript).

[1] Ning JG, Ren HL, Fang MJ. A constitutive mode

---

## [Decision Letter · Decision Letter 1]

PONE-D-25-16468R1A Macro-meso Damage Coupling Rock Mass Damage Model Based on Improved Internal Crack AnalysisPLOS ONE

Dear Dr. He,

Thank you for submitting your manuscript to PLOS ONE. After careful consideration, we feel that it has merit but does not fully meet PLOS ONE’s publication criteria as it currently stands. Therefore, we invite you to submit a revised version of the manuscript that addresses the points raised during the review process.

We look forward to receiving your revised manuscript.

Kind regards,

Kang Wang, Ph.D.

Academic Editor

PLOS ONE

Journal Requirements:

Reviewers' comments:

Reviewer's Responses to Questions

**Comments to the Author**

1. If the authors have adequately addressed your comments raised in a previous round of review and you feel that this manuscript is now acceptable for publication, you may indicate that here to bypass the “Comments to the Author” section, enter your conflict of interest statement in the “Confidential to Editor” section, and submit your "Accept" recommendation.

Reviewer #1: All comments have been addressed

Reviewer #2: All comments have been addressed

Reviewer #3: All comments have been addressed

2. Is the manuscript technically sound, and do the data support the conclusions?

Reviewer #1: Yes

Reviewer #2: Yes

Reviewer #3: Yes

3. Has the statistical analysis been performed appropriately and rigorously? 

Reviewer #1: Yes

Reviewer #2: Yes

Reviewer #3: Yes

4. Have the authors made all data underlying the findings in their manuscript fully available?

Reviewer #1: Yes

Reviewer #2: Yes

Reviewer #3: Yes

5. Is the manuscript presented in an intelligible fashion and written in standard English?

Reviewer #1: Yes

Reviewer #2: Yes

Reviewer #3: Yes

6. Review Comments to the Author

Reviewer #1: The authors have addressed all my comments and I have no further comments. This manuscript can be accepted.

Reviewer #2: (No Response)

Reviewer #3: (No Response)

7. PLOS authors have the option to publish the peer review history of their article (what does this mean? ). If published, this will include your full peer review and any attached files.

**Do you want your identity to be public for this peer review?** For information about this choice, including consent withdrawal, please see our Privacy Policy .

Reviewer #1: No

Reviewer #2: No

Reviewer #3: No

---

## [Author Response · Author response to Decision Letter 2]

2 Jun 2025

AUTHORS’ RESPONSE TO REVIEWERS’ COMMENTS

Journal: PLOS ONE

Title of the paper: A Macro-meso Damage Coupling Rock Mass Damage Model Based on Improved Internal Crack Analysis

Authors: Haian Liang Miao He Hongliang Zhao Yiru Zhang Qingrui Lu Shuren Hao Zhiying Gong Hao Guan

Manuscript ID: PONE-D-25-16468

Thank you for your letter and for the reviewers’ comments concerning our manuscript. Those comments are all valuable and very helpful for revising and improving our paper, as well as the important guiding significance to our researches. We have studied comments carefully and have made correction which we hope meet with approval.

Comment:

Point 1: The abstract can be further simplified, highlighting the core innovations and main conclusions of the research.

Response: Many thanks for your advice.

A concise and complete abstract is crucial for an article, as it can help readers quickly identify the main points of the article. Therefore, we have reorganized and rewritten the abstract, highlighting the core part of the research and the research results. (Page 1 lines 8-21 of the revised manuscript).

Point 2: Literature review: it is suggested that more international frontier research should be supplemented and the difference between this paper and existing work should be clarified.

Response: Many thanks for your advice.

We fully agree with your suggestion to increase international cutting-edge research by citing relevant articles in the article Added references [21,23,24,25,26] in the revised manuscript. and clarifying the differences between this article and existing research. (Page 4 lines 58-66 of the revised manuscript).

Point 3: It is suggested to explain the actual value of the model with a specific engineering Case.

Response: Many thanks for your advice.

If the established theory is combined with practical engineering, it will have a very positive effect on further enhancing the value of research. However, the study of internal elliptical joints is still mainly focused on the combination of indoor experiments, theoretical experiments, and numerical simulations [1,2,3,4,5], and there is not a lot of relevant engineering practice data. But, in the discussion section of this study, the value and limitations of using constitutive models in practical engineering were discussed. (Page 24 lines 321-346 of the revised manuscript). In future research, relevant engineering data will be collected and further validated to improve the practical applicability of the constitutive model, and attempts will be made to address the issues raised in the discussion section.

[1] Yang Y S, Cheng S P, Zhang Z R, et al. Study on the stress field concentration at the tip of elliptical cracks[J]. Reviews on advanced materials science, 2022, 61(1): 611-621.

[2] Niu Z, Wang X, Zhang L, et al. Investigation into the Failure Characteristics and Mechanism of Rock with Single Elliptical Defects under Ultrasonic Vibrations. Fractal and Fractional, 2024, 8(5): 261.

[3] Han Z, Li D, Li X. Experimental study on the dynamic behavior of sandstone with coplanar elliptical flaws from macro, meso, and micro viewpoints. Theoretical and Applied Fracture Mechanics, 2022, 120: 103400.

[4] Liu X R, Yang S Q, Huang Y H, et al. Experimental study on the strength and fracture mechanism of sandstone containing elliptical holes and fissures under uniaxial compression. Engineering Fracture Mechanics, 2019, 205: 205-217.

[5] Aliabadian Z, Sharafisafa M, Tahmasebinia F, et al. Experimental and numerical investigations on crack development in 3D printed rock-like specimens with pre-existing flaws. Engineering Fracture Mechanics, 2021, 241: 107396.

Point 4: It is necessary to clarify the existing research gaps and highlight the innovation of this paper.

Response: Many thanks for your advice.

We fully agree with your suggestion to clarify existing research gaps and highlight the innovation of this article. Therefore, we have added relevant content in the introduction section of the article to explain the shortcomings of existing research and the prominent characteristics of this study. (Page 3-4 lines 41-66 of the revised manuscript).

Point 5: It is necessary to further standardize academic expression and correct the problems of text errors, misuse of symbols and inconsistent formats.

Response: Many thanks for your advice.

Standardized grammar is the foundation of an article, so we have optimized the grammar expression and symbol usage throughout the text, unifying the format of the entire text.

---

## [Decision Letter · Decision Letter 2]

A Macro-meso Damage Coupling Rock Mass Damage Model Based on Improved Internal Crack Analysis

PONE-D-25-16468R2

Dear Dr. He,

We’re pleased to inform you that your manuscript has been judged scientifically suitable for publication and will be formally accepted for publication once it meets all outstanding technical requirements.

Kind regards,

Kang Wang, Ph.D.

Academic Editor

PLOS ONE

Additional Editor Comments (optional):

Reviewers' comments:

Reviewer's Responses to Questions

**Comments to the Author**

1. If the authors have adequately addressed your comments raised in a previous round of review and you feel that this manuscript is now acceptable for publication, you may indicate that here to bypass the “Comments to the Author” section, enter your conflict of interest statement in the “Confidential to Editor” section, and submit your "Accept" recommendation.

Reviewer #3: All comments have been addressed

2. Is the manuscript technically sound, and do the data support the conclusions?

Reviewer #3: Yes

3. Has the statistical analysis been performed appropriately and rigorously? 

Reviewer #3: Yes

4. Have the authors made all data underlying the findings in their manuscript fully available?

Reviewer #3: Yes

5. Is the manuscript presented in an intelligible fashion and written in standard English?

Reviewer #3: Yes

6. Review Comments to the Author

Reviewer #3: This paper introduces an improved macro-meso damage coupling model, demonstrating innovation in theoretical derivation and experimental validation. I agree with this article being published.

7. PLOS authors have the option to publish the peer review history of their article (what does this mean? ). If published, this will include your full peer review and any attached files.

**Do you want your identity to be public for this peer review?** For information about this choice, including consent withdrawal, please see our Privacy Policy .

Reviewer #3: No

---

## [Editor Report · Acceptance letter]

PONE-D-25-16468R2

PLOS ONE

Dear Dr. He,

I'm pleased to inform you that your manuscript has been deemed suitable for publication in PLOS ONE. Congratulations! Your manuscript is now being handed over to our production team.

Kind regards,

on behalf of

Dr. Kang Wang

Academic Editor

PLOS ONE